# Social Capital—Can It Weaken the Influence of Abusive Supervision on Employee Behavior?

**Jie Cheng [1], Myeong-Cheol Choi [1],\* and Joeng-Su Park [2],\***

[1]  Department of Business, Gachon University, Seongnam 13120, Republic of Korea
[2]  Department of Global Management, Chonnam National University, Yeosu 59626, Republic of Korea
\*   Correspondence: oz760921@gachon.ac.kr (M.-C.C.); joengsu@jnu.ac.kr (J.-S.P.)

**Abstract:** The travel industry has been severely affected by the COVID-19 outbreak. The operating pressure on enterprises has sharply increased, leading to the prominent phenomenon of abusive supervision. Managers employ this management method so that employees perceive work pressure as motivation to work harder and improve their performance. Employees may adopt the behavior of defensive silence to protect themselves from abusive supervision, which can subsequently affect employee behavior. However, social capital and relationships may lessen this effect. This study analyzed survey data on 475 workers from the Chinese tourism service industry to examine the mediating role of workers' defensive silence under abusive supervision, employee behavior, and the moderating role of social capital. The results showed that abusive supervision does not promote employee performance but hinders employee growth. Employees' defensive silence also affects employee behavior and has a partially mediating role in the relationship between abusive supervision and employee behavior. Social capital can mitigate the negative impact of abusive supervision on employee behavior. This study theoretically expands the applicable scope of employee silence as a mediating variable and social capital as a moderating variable. It is helpful for managers to change their negative leadership style, follow the suggestions of employees, pay attention to the organizational atmosphere, and enhance their team cohesion.

**Keywords:** abusive supervision; employee silence; service-oriented organization citizenship behavior; social capital





## 1. Introduction

Following the COVID-19 outbreak, the Internet and e-commerce industries took advantage of the potential development. However, traditional industries were greatly impacted by COVID-19, especially tertiary industries such as tourism. According to the World Tourism Organization [1], a surge in global COVID-19 infections led to travel restrictions across 98% of tourist destinations. The pandemic has had a heterogeneous impact on the business performance of tourism enterprises, and the overall development of tourism has been sluggish [2]. In the business environment, the pressure faced by enterprise managers has increased, leading to an increase in the phenomenon of abusive supervision. In situations of intense pressure, managers can employ the practice of abusive supervision while dealing with employees. On the one hand, these managers may be under high pressure, leading to the loss of emotional control. On the other hand, managers may believe that this management method can stimulate employees to work harder and provide better performance [3].

It is worth noting that abusive supervision is subjective, and employees can have a subjective evaluation of the leader's behavior of abuse. Some employees may take the leader's behavior to be insulting, hurtful to their self-esteem, and causing psychological harm. But some employees will perceive such management activities as just normal. People in different cultures will have a different understanding of phenomena such as leadership

style, performance, and creativity [4]. For example, Chinese employees have fewer negative reactions compared with American employees [5].

Abuse and criticism can undermine the dignity of employees. In the short term, employees may try to improve their work performance to prove themselves and reduce the degree of abuse inflicted by their leaders [6]. However, in the long run, this has a significantly negative impact on the career development of employees and the cohesion of the enterprise. This is not conducive to the sustainable development of enterprises. Abusive supervision is seen as a major cause of employees' behavior of silence [7]. At work, employees have more opportunities to identify problems within the enterprise as early as, if not sooner than, their leaders. If the employees convey their opinions, the organization can optimize the decision of the leader by seizing the opportunity. However, in reality, employees often choose to be silent, especially if the cultural situation involves a high level of power inequality, such as in China, South Korea, and other countries that are influenced by Confucian culture. The cultural conditions encourage employees to remain silent and follow the "silence is gold" rule. In certain situations, managers cannot obtain key information in time, which negatively affects enterprise performance [8].

Continuous abusive supervision can result in emotional exhaustion, a reduction in social activities and a negative impact on other employee behaviors. This will affect organizational citizenship behavior (OCB) [9–12] and increase counterproductive work behavior (CWB). In the tourism service industry, the employees, along with their colleagues and leaders, have direct contact with the customers while providing their services. Accordingly, enterprises motivate employees' service-oriented organizational citizenship behavior (SOCB) to improve service performance.

Against this background and pressurized by an industry that has been severely affected by the pandemic, employees' increased exposure to abusive supervision has only become worse. Therefore, whether the pressure caused by a negative leadership style or ego depletion resulting from employees' silence will affect employees' SOCB is a problem that needs to be studied in depth.

The study of social capital has become popular in recent years. In the context of the current study, social capital involves interpersonal relationships in the workplace. In addition to facing leaders, employees also communicate with their colleagues in the workplace. A harmonious relationship between colleagues affects employees' behavior to some extent and may also influence the relationship between employees and leaders. In an environment in which the management's supervision is abusive, if the relationship between colleagues is harmonious, it can reduce the negative emotions of employees. In addition, relational capital promotes knowledge sharing [13]. Therefore, employees who do not give advice because of the leader can do so for the common growth of the team. Employees who are silent in the presence of leaders may discuss the enterprise's problems and solutions with their colleagues. Employees who want to leave the workplace because of strict management may choose to stay for their colleagues. Therefore, whether and how the relationship between colleagues, that is, relational social capital, moderates the relationship between employees and leaders in China's tourism industry needs to be deeply investigated and verified.

Although there are some studies on the influence of social capital, for example, the study that found that social capital can alleviate an individual's propensity towards knowledge hiding [14], no studies have explored the moderating effect of social capital. Therefore, this study is an innovative exploration.

## 2. Literature Review

### 2.1. Abusive Supervision

The concept of abusive supervision, proposed by Tepper [15,16], refers to the continuous verbal or nonverbal hostile behavior perceived by subordinates and inflicted by managers. It must be noted that it does not include physical contact. Abusive supervision has recently become a popular topic in research on leadership styles in the field of organiza-

tional behavior. However, there is a lack of comprehensive understanding of the topic [17]. Studies on the impact of abusive supervision on employees' behavior have shown that subordinates deal with the sense of injustice caused by humiliation and abuse by reducing their own resources. For example, employees will reduce their pro-organizational behavior [18], OCB [9–12], and voice behavior [19], consequently depreciating their work performance [9,20,21]. Mitchell and Ambrose [22] have found that employees suffering from abusive supervision may choose to directly retaliate against their supervisors, causing supervisor-oriented workplace bias behavior, or motivate their colleagues to retaliate against the organizations' members or other colleagues, inducing organization-oriented and interpersonal-oriented workplace bias behaviors.

### 2.2. Employee Silence

Morrison and Milliken [23] (2000) were the first to define employee silence and believe that silence is a collective behavior. They have suggested that employees retain their personal views on the potential problems with the organization. Zheng et al. [24] believe that employees' silent behavior is more common in enterprises with high power inequality, collectivism, and interpersonal orientations. They conducted indigenous studies in China, and employee silence behavior was divided into the following three dimensions: acquiescent silence, defensive silence, and disregardful silence. A negative leadership style, such as abusive leadership, promotes employee silence and reduces the voice of the employee [25]. A harmonious relationship among colleagues enables employees to actively express their views and be willing to reach a consensus through communication. To maintain a good relationship with their colleagues, employees will selectively express their opinions or remain silent [26,27].

For example, similar to Andersen's fairytale titled "The Emperor's New Clothes", when the employees stay silent and do not give feedback, managers assume that the enterprise is operating well and lose out on vital opportunities for development [28]. However, some scholars believe that employee silence can avoid internal conflicts within the organization, help maintain interpersonal harmony, and improve the quality of team collaboration [29]. Thus, silence can also be valuable to employees and organizations [30].

### 2.3. Service-Oriented Organization Citizenship Behavior (SOCB)

Bettencourt et al. [31] were the first to propose the concept of Service-oriented Organization Citizenship Behavior in their research on behaviors in the service industry. They defined SOCB as the spontaneous behavior of the individual in front-line service, which goes beyond the regulations but is beneficial to service performance. They divided the SOCB into the following three dimensions: loyalty, engagement behavior, and service provision. Previous research has shown that organizational service performance in the hotel industry has a positive impact on SOCB [32]. Colleagues' SOCB can influence other employees, prompting them to partake in SOCB, subsequently promoting the enterprise's service performance [33]. A positive leadership style promotes employees' SOCB [34,35]. A small number of studies have explored the impact of negative leadership styles on the SOCB of employees. For example, Lyu [36], using the conservation of resources (COR) theory, found that abusive supervision can decrease the work engagement of employees, negatively affecting their SOCB in the service industry.

### 2.4. Social Capital

As for the definition of social capital, scholars represented by Bourdieu believe that "Social capital is the aggregate of the actual or potential resources" [37]. It should be made clear that with Bourdieu, social capital refers to power relations, meaning that everything relates to power, as in having something that someone does not have or knowing an individual that someone else does not know; it is not just a sum [38]. With the deepening of research, the definition of social capital has become more biased towards its existence

in the relationships among people [38–41]. Coleman believes that "Social capital, comes about through changes in the relations among persons that facilitate action" [38].

This study is more in favor of Tsai and Nahapiet's proposition [39–41]. From their viewpoint, social capital was used to describe relational resources, embedded in cross-cutting personal ties. Nahapiet and Ghoshal [40,41] divided social capital into three dimensions according to different attribute clusters: structural, relational, and cognitive dimensions. This study refers to the dimensions they divided and focuses on the content of relational resources and relational capital. We concentrate on interpersonal relationships and consider social capital as the human capital that individuals or organizations acquire through their friends, colleagues, or social membership. These include attributes such as respect, trust, friendship, and recognition.

Research on the concept of social capital is relatively mature, but the research on the impact of social capital is not rich enough. Studies have shown that organization culture has an impact on social capital [42]. In the tourist business, social capital will affect the turnover intention of hotel employees [43]. At present, few studies have explored social capital as a mediating variable or a moderating variable.

## 3. Theory and Hypotheses

### 3.1. Abusive Supervision and Employee Silence

From the employees' perspective, abusive supervision is a threatening stressor. Employees who are subjected to abusive supervision on a regular basis report feeling more stressed and emotionally worn out, according to Breaux et al. [44]. These workers experience emotional weariness and a lack of control, which frustrates them [9]. Similar studies have found that abusive supervision by seniors leads to depression and anxiety among junior employees [18,45].

COR theory points out that when employees face threats from the outside world, they lose their internal resources, including their emotional resources. If employees do not receive support from their leaders and organizations, they may not share their knowledge [46]. In order to prevent resource loss brought on by psychological contract violations, bullied employees also develop knowledge-concealing behaviors [47]. Similar to workplace bullying, continuous abusive supervision may also cause employees to protect their remaining resources from further loss and consequently exhibit less knowledge-sharing behavior [48]. Thus, fear strengthens the link between abusive supervision and employees' defensive silencing of themselves [7].

According to the social exchange theory, when a superior such as a leader adopts abusive supervision, the quality of the exchange relationship between the superiors and subordinates diminishes [12,49]. In negative exchange relationships, employees are less likely to give advice [50].

Additionally, when employees face abusive supervision, they adopt silencing behaviors and reduce social exchange to avoid the continuous exhaustion of emotional resources. According to the definition of employee silence and its dimensions, this type of employee silence can be categorized as defensive silencing. Therefore, this study raises the following hypothesis:

**Hypothesis 1 (H1).** *Abusive supervision has a positive effect on employee silence.*

### 3.2. Abusive Supervision and SOCB

Abusive supervision in the workplace has many negative effects, including on OCB [51]. People facing abusive supervision demonstrate less OCB [52]. Abusive supervision is a chronic stressor that leads to the elimination of the resources needed to achieve goals [53]. When faced with constant criticism and ridicule, employees need to overcome stress to use their cognitive and emotional resources [54].

Reducing OCB protects resource balance when workers are exposed to unfavorable leadership methods, in accordance with COR theory. Employees must spend time and effort using SOCB as an internal resource. According to studies, employees' efforts to provide good customer service may decrease [55]. According to social exchange theory, abusive supervision reduces employees' reciprocity towards their leaders or organizations; moreover, they develop a negative attitude to deal with the low-quality exchange connection [56]. If the organization and leaders maintain a high-quality social exchange relationship with their employees, employees will provide SOCB as a resource exchange. When employees feel that the organization is reasonable and fair, they will have a positive work attitude and perceived behavioral control, which will stimulate spontaneous SOCB [57].

From the perspective of COR and social exchange theory, employees' SOCB toward customers in the tourism service industry leads to high-quality social exchange resources. However, when suffering from abusive supervision, employees will feel a weakened sense of self-control and will not perform actions that are beneficial to the organization. Accordingly, they will reduce the quality of their social exchanges to preserve existing resources. Therefore, it can be inferred that, in the tourism service industry, abusive leaders reduce the SOCB of their employees. This study thus presents the following hypothesis:

**Hypothesis 2 (H2).** *Abusive supervision has a negative effect on SOCB.*

### 3.3. Employee Silence and SOCB

Baumeister [58] was the first to propose the ego depletion theory, which could explain the changes in individual psychological and organizational behavior caused by resource loss. In terms of organizational behavior, ego depletion reduces input [59], work output [60], and OCB [61,62].

Most studies have shown that admonition behavior has positive effects on employees and organizations, and employee silence has negative effects on employees and organizations. Employees' silence affects their cognition and emotions, and emotions are an important factor in regulating behavior. Emotional environments or events affect individuals' creativity and trigger negative emotions and silencing behavior among employees. Such negative emotions are not conducive to stimulating the creativity of employees [63]. Employees' silence makes them indifferent toward [64] OCB and creativity, both of which consume employees' internal resources. Employees' silence also creates a conservative atmosphere that negatively affects their OCB.

Accordingly, in the tourism service industry, SOCB indicates if the employees are good at expressing themselves and consuming internal resources to actively provide excess services. Therefore, if employees become accustomed to working silently, it is difficult for them to take the initiative and provide high-quality services. Therefore, it can be inferred that employee silence is not conducive to the SOCB of employees in the tourism service industry. Thus, the following hypothesis is proposed:

**Hypothesis 3 (H3).** *Employee silence has a negative effect on SOCB.*

### 3.4. Mediating Role of Employee Silence

The negative impact of employee silence on the organization is self-evident. Moreover, research on the mediating role of employee silence is not sufficiently systematic and comprehensive. Studies have confirmed that there is an intermediary relationship between employees' silence, negative gossip in the workplace, and employee innovation and performance [65]. Defensive silencing has a mediating role in the relationship between workplace rejection and interpersonal bias [66]. Wang et al. (2020) conducted a study using employees of a large hotel in Taiwan. They found that employee silence mediates the connection between abusive supervision, work engagement, and job satisfaction [67]. Under abusive supervision and task performance, defensive silence has a mediating effect [68]. According

to COR and the social exchange theory, in contexts with high power inequality, such as in China, when employees suffer from abusive supervision, they choose to stay silent, hide, or selectively express their views. Thus, abusive supervision makes employees feel that they are not valued and have no right to speak, and they will eventually lower the OCB to act as a vent or resistor. At the same time, according to the ego depletion theory, employee silence is regarded as self-loss. It can be inferred that employee silence plays a mediating role between abusive supervision and SOCB of the employees in the service industry.

According to this line of reasoning and given the context of this study, it can be inferred that employees in the tourism service industry in China first maintain silence at work and then reduce their SOCB. Therefore, we propose the following hypothesis:

**Hypothesis 4 (H4).** *Employee silence mediates the relationship between abusive supervision and SOCB.*

*3.5. Moderating Role of Social Captial*

There are few studies on the moderating role of social capital. According to the literature review and relevant definitions, it can be considered that social capital is a resource, which can be explained using the COR theory. High-quality workplace relationships bring resource benefits to employees. This is because tacit understanding and efficient communication between colleagues reduces work stress and tension among employees [69].

The social network has a considerable and advantageous impact on information exchange, and social capital plays a crucial role in its promotion [70]. Internal social capital has a moderating effect on the relationship between leader–member exchange (LMX) and job creation [71]. Abusive supervision reflects the behavior of the leader. If the overall organizational culture and working atmosphere of the enterprise are sufficiently harmonious, frequent contact between colleagues will reduce the negative emotions of the employees that have been subjected to abusive supervision. Workplace friendships mediate the impact of workplace bullying on employee silence [72]. Good workplace interpersonal relationships will bring resource benefits to employees. This relational social capital resulting from the relationships between colleagues can make up for the resource loss of employees in other aspects. To some extent, this sum of resources can be maintained.

Social capital can be explained by the social exchange theory as well. Good interpersonal workplace relationships enable employees to gain greater organizational support and trust. Therefore, employees will have a stronger social willingness to exchange and share knowledge, resulting in high-quality reciprocal exchanges and less silent behavior. This may neutralize the low-quality exchange relationships caused by abusive supervision. Based on the above perspective, the following hypotheses are proposed:

**Hypothesis 5 (H5).** *Social capital has a negative moderating role in the relationship between abusive supervision and employee silence. The stronger the social capital, the weaker the effect of abusive supervision on employee silence.*

**Hypothesis 6 (H6).** *Social capital has a negative moderating role in the relationship between abusive supervision and SOCB. The stronger the social capital, the weaker the effect of abusive supervision on SOCB.*

The above hypotheses can be summarized as shown in the figure below. Figure 1 shows the research model.

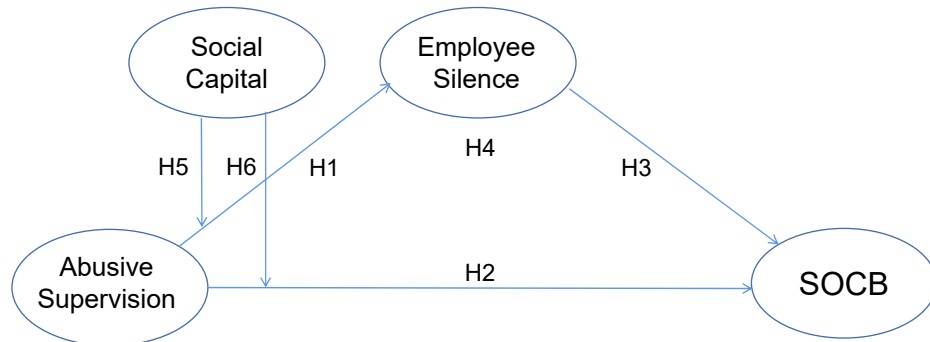

**Figure 1.** Research model.

## 4. Methodology

We used SPSS 25.0 for data analysis with Bootstrap. AMOS 25.0 was used for confirmatory factor analysis (CFA) and structural equation modeling (SEM).

### 4.1. Measures

Abusive supervision was quantified using the measurement questionnaire developed by Tepper et al. [15], which included five questions, such as "My boss told me that my thoughts and feelings were stupid." and "My boss made a fool of me in public.", etc.

Employee silence was measured using the dimension of "Defensive Silence", which was developed by Zheng [24] and contained a total of five items, such as "There was little chance that the leader would take my advice, so I will not tell others my ideas." and "There's no need to offend your boss or colleagues by saying something.", etc.

SOCB, or the "service delivery" dimension, was measured using the scale developed by Bettencourt et al. [31] and had six items, such as "In any case, I will be very kind to customers and offer to help." and "I can respond quickly to customer requests, even after hours.", etc.

Social capital, or the "social relationship", was measured using the scale developed by Gu et al. [73] and consisted of five questions, such as "My colleagues and I trust and promote each other at work." and "My colleagues and I shared our team vision and defined our goals.", etc.

The aforementioned scales were tested for high reliability, validity, and general applicability. The present study used the Likert five-point scale, ranging from 1 = Strongly Disagree, 2 = Disagree, 3 = Neutral, 4 = Agree, and 5 = Strongly Agree, to assess the relationship among abusive supervision, employee silence, SOCB, and social capital.

This study refers to scales translated by Chinese scholars and used in China. Other English scales still need to be translated. In order to ensure the accuracy of the translation and avoid any ambiguity, we invited two experts who are proficient in both languages to assist in the translation. With the help of professors, the differences between the Chinese and English scales were compared, and the prototype of the questionnaire was formed after modification and adjustment. To guarantee the questionnaire's validity, small sample tests and interviews were conducted first, and then some sentences were adjusted to make the questionnaire more suitable for the Chinese context. Finally, a formal questionnaire survey was conducted.

### 4.2. Sample and Procedures

This study selected employees from China's tourism service industry as the research sample. In Shanghai, Nanjing, Hangzhou, and other Chinese cities, the study materials were dispersed among tourist attractions, hotels, travel agencies, and other businesses. Due to the outbreak, the questionnaire was distributed online and ran from March to May 2022. We sent questionnaires to their employees through the administrator of these companies. The employees were informed in advance that the information was completely

confidential and would only be used for scientific study. After the survey distribution, 518 questionnaires were collected, among which questionnaires with poor responses were deleted, and finally, 475 questionnaires were used for the study. The survey had a 91.70% response rate.

*4.3. Control Variable*

Previous studies controlled the gender, age, education background, and job position of participants [74,75]. This information may affect the variable relationships. Hence, in this study, we also controlled for the gender (gender was coded as a dummy variable, 0 for women and 1 for men), age, education background, and work experience of respondents.

## 5. Results

*5.1. Descriptive Analysis*

According to the data recovery of 475 samples, the descriptive statistical analysis of the demographic variables showed that the sample structure conformed to the industry rules. Specific data information is shown in Table 1.

**Table 1.** Descriptive Analysis of Sample.

| Demographic Variable | Type | Frequency | Ratio |
|---|---|---|---|
| Sex | Male | 180 | 37.89% |
| | Female | 295 | 62.11% |
| Age | Age under 25 years | 207 | 43.58% |
| | 26–30 Years old | 112 | 23.58% |
| | 31–35 Years old | 72 | 15.16% |
| | 36–40 Years old | 53 | 11.16% |
| | 41–50 Years old | 21 | 4.42% |
| | Age more than 50 years | 10 | 2.11% |
| Education Background | High school degree or below | 11 | 2.32% |
| | College degree | 281 | 59.16% |
| | Bachelor degree | 155 | 32.63% |
| | Master's degree | 19 | 4.00% |
| | PhD degree or above | 9 | 1.89% |
| Tenure | Work for 1–5 years | 289 | 60.84% |
| | Work for 6–10 years | 91 | 19.16% |
| | Work for 11–15 years | 66 | 13.89% |
| | Work for 16–20 years | 10 | 2.11% |
| | Over 20 years of work | 19 | 4.00% |
| Position | General staff | 257 | 54.14% |
| | Low-level managers | 103 | 21.60% |
| | Middle management | 60 | 12.72% |
| | Top management | 32 | 6.66% |
| | Others | 23 | 4.88% |
| Type of Job | Service staff | 228 | 47.93% |
| | Marketing/Advertising | 94 | 19.82% |
| | Administrative/HR/Accounting | 81 | 17.02% |
| | Technology/R & D | 20 | 4.14% |
| | Others | 52 | 11.09% |

*5.2. Model Validation Test*

In this study, Cronbach's $\alpha$ was used to test the scales by employing the software SPSS 26. The results showed that the Cronbach's $\alpha$ of each variable was greater than 0.8 (Table 2), indicating that the scale had very good internal consistency and high reliability. Moreover, deleting any question would have reduced the reliability of the original scale, indicating that there were no redundant items. Therefore, we retained all the questions and maintained good consistency with the scales.

**Table 2.** Descriptive statistics and correlation analysis.

|  | Mean | SD | AVE | ABS | DS | SOCB | RSC |
|---|---|---|---|---|---|---|---|
| ABS | 2.55 | 0.999 | 0.506 | 0.836 | | | |
| DS | 2.68 | 0.992 | 0.565 | 0.463 ** | 0.863 | | |
| SOCB | 4.02 | 0.628 | 0.599 | −0.507 ** | −0.573 ** | 0.898 | |
| SC | 3.91 | 0.712 | 0.616 | −0.149 ** | −0.330 ** | 0.221 ** | 0.888 |

NOTE: The diagonal is the Cronbach's α of variables; the lower triangle is the Pearson correlation coefficient between variables; ** $p < 0.01$. ABS—abusive supervision; DS—employee silence; SOCB—Service-oriented Organization Citizenship Behavior; SC—social capital.

Confirmatory factor analysis was performed for each study variable using the statistical software AMOS 26. The results of the model fit were: $\chi^2/df$ = 1.158, less than 3; GFI = 0.959, AGFI = 0.948, and CFI = 0.994, all greater than 0.9; and SRMR = 0.0288, RMR = 0.030, and RMSEA = 0.018, all less than 0.05. Overall, the model used in this study had a good model fit.

Convergent validity refers to the degree to which different measurements of the same latent variable can have a common factor. The degree of model convergence can be mainly judged by the payload of each variable factor, average variance extraction rate AVE (average variance extracted), and combined reliability CR (construct reliability) [76,77]. The analysis results showed that the factor loads corresponding to the study variables were greater than 0.5, AVE values were greater than 0.5, and CR values were greater than 0.7 (Table 2). This indicated that the measurement scale of each variable in this study had an ideal aggregate validity.

The discriminant validity was assessed using the square root of the AVE and the comparative size of the correlation coefficients between the two variables [76]. This study had a good correlation between the variables, which provided a reference for further testing of the model and hypotheses. The AVE square root of each variable was greater than the absolute value of the correlation coefficient between a given variable and the other variables (Table 2). Therefore, the discriminant validity between the variables was good.

To avoid the common method bias (CMB) resulting from the questionnaire [78], we performed a CMB test using Harman's single factor test. An exploratory factor analysis (EFA) of the study sample data was performed using the software SPSS 26. The principal factor analysis provided four factors that were greater than one; the first factor explained 19.012% of the total variance and less than 40% of the cut-off value. This indicated that, in this study, no single factor explained the majority of the variance [79].

*5.3. Hypotheses Testing*

In this study, the study model was hypothesis tested using SEM and AMOS 26 software. The research model is shown in Figure 2. Table 3 shows the estimates and regression results.

**Table 3.** SEM Analysis.

|  | Path | Unstandardized Estimates | Standardized Estimate | S.E. | C.R. | | p |
|---|---|---|---|---|---|---|---|
| H1 | ABS → DS | 0.523 | 0.517 | 0.056 | 9.295 | | *** |
| H2 | ABS → SOCB | −0.172 | −0.302 | 0.033 | −5.200 | | *** |
| H3 | DS → SOCB | −0.272 | −0.484 | 0.037 | −7.309 | | *** |
| H5 | ABS × SC → DS | −0.418 | −0.219 | 0.058 | −4.144 | | *** |
| H6 | ABS × SC → SOCB | −0.054 | −0.050 | 0.030 | −1.069 | | 0.283 |
| | | | | | Lower | Upper | |
| H4 | ABS → DS → SOCB | | −0.250 | 0.038 | −0.329 | −0.182 | *** |

NOTE: *** $p < 0.001$. ABS—abusive supervision; DS—employee silence; SOCB—Service-oriented Organization Citizenship Behavior; SC—social capital.

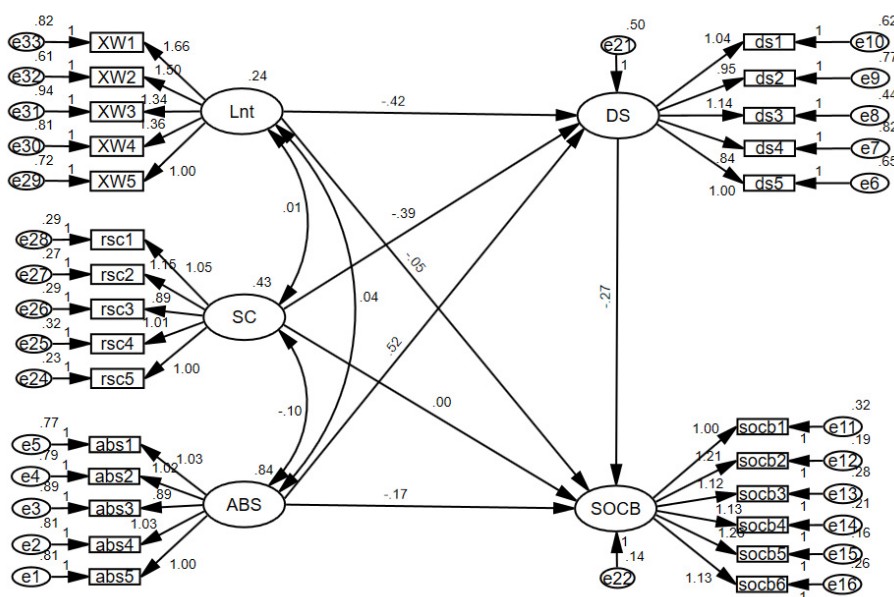

**Figure 2.** Research model. NOTE: ABS—abusive supervision; DS—employee silence; SOCB—Service-oriented Organization Citizenship Behavior; SC—social capital; Lnt = ABS × SC, Interactive Item.

To further check the value of the mediation effect of employee silence, the PROCESS macro for SPSS was used for the mediation effect of employee silence. Model 4 was selected, and the confidence interval was set to 95%. The specific results are shown in Table 4.

**Table 4.** Mediation effect of employee silence (Bootstrap).

| Independent Variable | Dependent Variable | Type of Influence | Effect Size | Boot LLCI | Boot ULCI | Effect Ratio |
|---|---|---|---|---|---|---|
| Abusive Supervision | SOCB | Total Effect | −0.3191 *** | −0.3680 | −0.2701 | |
| | | Direct Effect | −0.1936 *** | −0.2432 | −0.1440 | 60.67% |
| | | Indirect Effect | −0.1255 *** | −0.1597 | −0.0938 | 39.33% |

NOTE: *** $p < 0.001$.

The Bootstrap test showed that the *p*-values for the total and direct effects were less than 0.001. The table shows the indirect effect value of employee silence on abusive supervision and employee SOCB, which was −0.1255 (LLCI = −0.1597, ULCI = −0.0938, interval excluding 0), accounting for 39.33% of the total effect value of −0.3191. The direct effects accounted for 60.67% of the total effect. The data further proved that employee silence played a partially mediating effect on the relationship between abusive supervision and employee SOCB. To further verify the moderating effect and whether the hypotheses in this study hold, another 5000 Bootstrap tests were performed using the SPSS 26 Process. Model 1 was selected with a 95% confidence interval. The specific results are presented in Table 5.

**Table 5.** Moderation effect of social capital (Bootstrap).

| Independent Variable | Dependent Variable | Effect Size of Lnt | SD | Boot LLCI | Boot ULCI |
|---|---|---|---|---|---|
| Abusive Supervision | DS | −0.2360 *** | 0.0504 | −0.3350 | −0.1370 |
| | SOCB | 0.0428 | 0.0327 | −0.0214 | 0.1070 |

NOTE: *** $p < 0.001$.

According to the above table, the moderating effect of social capital and the interaction between abusive supervision and employee silence was −0.2360 (LLCI = −0.3350, ULCI = −0.1370, interval excluding 0). The data showed that H5 was also supported. The

mean and standard deviation of the variables were drawn to further test the moderating variable [80–82], as shown in the results in Figure 3.

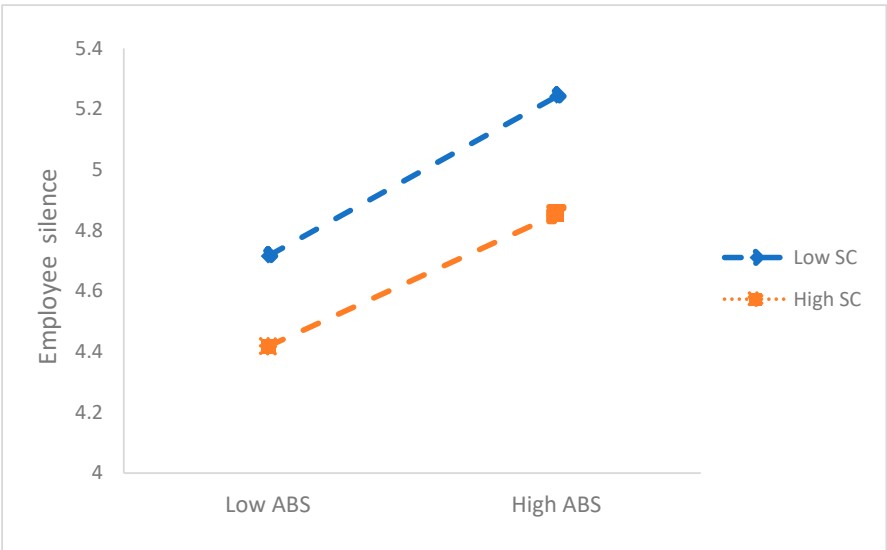

**Figure 3.** Moderation effect of SC between ABS and DS. NOTE: ABS—abusive supervision; DS—employee silence; SC—social capital.

The moderating effect of social capital on the relationship between abusive supervision and SOCB was 0.0428 (LLCI = −0.0214, ULCI = 0.1070, and the interval contained 0). H6 was not supported.

## 6. Conclusions

### 6.1. Discussion

This study analyzed the effect of abusive supervision on employee silence and SOCB in Chinese tourism companies, and the mediating effect of employee silence and the moderating effect of social capital. The results of this study are as follows.

First, abusive supervision does not promote employee performance but hinders employee growth. Compared with Western countries, in countries such as China and South Korea, there is higher power inequality. Furthermore, corporate culture is influenced by Confucian culture. The authority of leaders is more powerful, and the culture of predecessors and successors is prevalent. The phenomenon of abusive supervision in these countries' enterprises is not rare. The employees who have suffered abusive supervision may learn and imitate abusive behavior toward their subordinates and successors [83]. Such managers often think that the employment of pressure management can stimulate the potential of employees, motivate them, and improve their ability to resist pressure. They believe that the direct high-pressure management method is effective for employees [6]. However, can this stressor stimulate employee potential to improve performance? The answer is no. The results of this study showed that employees are not motivated by abusive supervision, disproving that "beating shows affection and scolding shows love." Abusive supervision affects employees' emotions and cognition. It leads to a sense of lack of organizational support, organizational trust, and work security among employees, and they fall into self-denial. Abusive treatment also affects employees' work behavior, negatively affects their behaviors that are conducive to the organization, and even results in behaviors that are harmful to the organization [9–12].

Second, employee silence affects employee behavior and partially mediates the relationship between abusive supervision and employee behavior. In this study, employee silence is regarded as a special employee behavior. Maintaining silence seems to be an act of self-protection. However, it leads to internal emotional consumption, which has a further significantly negative impact on the employees [28]. In the tourism service industry, SOCB

indicates whether employees are good at expressions, active performance, and consuming internal resources to actively provide excess services. For employees who are accustomed to working in silence, it is difficult to demonstrate initiative and provide good service. Accordingly, the silent employees' SOCB declines. Alternatively, the abused employees will choose to remain defensively silent but show their dissatisfaction through their work behavior. Therefore, one behavior influences another. The mediating role of silence is evident in this case.

Third, social capital can mitigate the negative effects of abusive supervision on employee behavior. According to H5, social capital negatively moderates the relationship between abusive supervision and employee silence. The stronger the social capital, the weaker the effect of abusive supervision on employee silence. According to the moderating effect results, when social capital is high, the effect of abusive supervision on employee silence is greatly reduced; therefore, the slope is also significantly flat. This shows that employees tend to automatically rationalize negative leadership behavior. This type of relational social capital results from the relationships between colleagues and can compensate for the resource loss of employees in other aspects to a certain extent. Accordingly, the sum of resources can be maintained. For organizational problems that abused employees are not willing to report to the leaders, they may confide in their colleagues and share their ideas [13].

H6 was not supported, and the reasons are as follows: First, when the measurement scale was revised during the translation process, there may have been a semantic understanding bias, and situations may not have been applicable. Second, at the time of sample collection, the study subjects were employees from the Chinese tourism service industry. However, some of the participants may not directly face customers and provide services and instead may be engaged in other activities such as financial accounting and Internet operations. This may have led to some errors in the data. Future studies can expand the sample size, refine the type of research objects, and avoid such errors that could affect the analysis results. Third, according to the theoretical considerations taken by this study, the hypotheses concerned the following four groups: employees, leaders, colleagues, and customers. Here, employees, leaders, colleagues, and customers are separate components. The abusive supervision of employees by leaders reduces the SOCB of employees when they deal with customers; the friendly relationship between colleagues cannot alleviate the impact between the other two groups, and the social exchange relationship of employees is difficult to exchange between leaders, customers, and colleagues, which is too complicated. Therefore, the hypothesis was not supported.

In terms of theoretical contribution, this study explains the motivation and origin of abusive supervision through the exploration of Confucian cultural thought, particularly from an industry perspective. This thought enriches the influence mechanism between abusive supervision and employee behavior especially in corporate settings. Our study expands the research on the influencing factors of employees' defensive silence. Furthermore, it reveals the mediation pathway of employee silence. It also expands the applicability of social capital as a moderating variable and makes theoretical contributions to the study of relational social capital.

In terms of practical implications, deepening managers' understanding of abusive supervision and its effect on the tourism service industry in the context of Chinese Confucian culture is conducive to managers changing their ineffective management behaviors. It helps managers realize the harm caused by employee silence to the organization and motivates them to establish smooth channels of advice. It helps managers pay attention to the organizational atmosphere, strengthen the relationship management between colleagues, and enhance a sense of teamwork.

*6.2. Future Research and Limitations*

The self-report method and a corresponding scale were used in this study. The scales developed for these variables were investigated from the perspective of employees,



while the consequences were mostly considered from the perspective of employees. In future studies, the measurement scales of leadership and employee mutual evaluations can be used. Organization-level and individual-level reports can be used to obtain more objective data. Cross-level research methods can be used to explore the relationship between leadership styles and employee behavior.

In future studies, longitudinal research methods can be adopted to collect data at different times and explore the influence of different times through multiple cross-sections.

This study is limited to the national and industry contexts. At present, the model only considers a sample of employees from China's tourism service industry, and the conclusion may not be universal. Future studies can explore whether the theoretical model can be established in other countries and industries. Comparative studies can also be conducted on data from different situations to explore the differences in the effects of situations.

**Author Contributions:** J.C. is responsible for the conceptualization, methodology, resources, and writing—original draft preparation. M.-C.C. is responsible for writing—review and editing and supervision. J.-S.P. is responsible for the formal analysis, investigation, and methodology. All authors have read and agreed to the published version of the manuscript.

**Funding:** This research received no external funding.

**Institutional Review Board Statement:** Survey studies in business administration are exempt from review and approval in Gachon Univ. Ethical review and approval were waived for this study, due to school policy.

**Informed Consent Statement:** Informed consent was obtained from all subjects involved in the study.

**Data Availability Statement:** Not applicable.

**Conflicts of Interest:** The authors declare no conflict of interest.

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
