# Peer review of "Social Capital—Can It Weaken the Influence of Abusive Supervision on Employee Behavior?"

_sustainability, doi:10.3390/su15032042_

Round 1
Reviewer 1 Report
Title: Social capital–can it weaken the influence of abusive supervision on employee behavior?1. It is recommended that the contribution of this paper be mentioned in the abstract. 2. At the beginning of the introduction, it is suggested to update the literature on social capital and abusive supervision to 2022, and need to define the problem in a broader context.The innovation of this paper can be compared with the achievements of other scholars to highlight the contribution and innovation of this paper.
3. It is suggested that add the hypothesis (H1-h6) to Figure 1, which will make readers more clearly understand the association and mechanism between them. 4. In section of Methodology, It is recommended to add specific research models. 5. Policy implication should be consistent with results and models.
Author Response
1.It is recommended that the contribution of this paper be mentioned in the abstract.
=>These are added in the abstract.
- At the beginning of the introduction, it is suggested to update the literature on social capital and abusive supervision to 2022, and need to define the problem in a broader context. The innovation of this paper can be compared with the achievements of other scholars to highlight the contribution and innovation of this paper.
=>These are added in the introduction.The literature has been updated.
- It is suggested that add the hypothesis (H1-h6) to Figure 1, which will make readers more clearly understand the association and mechanism between them.
=>These have already been added. Figure 1 has been updated.
- In section of Methodology, It is recommended to add specific research models.
=>Research models have already been added.
- Policy implication should be consistent with results and models.
=>The content of Practical implication is proposed according to the model and the results.
- Can be improved:Are all the cited references relevant to the research?
=>It has been improved. Had already improved.
- Can be improved:Are the research design, questions, hypotheses and methods clearly stated?
=>It has been improved. More detailed instructions have been added.
- Can be improved:Are the conclusions thoroughly supported by the results presented in the article or referenced in secondary literature? =>It has been improved. The conclusion is the author's analysis and thinking based on the validation results. Added some literature to support it.

Reviewer 2 Report
Minor language editing is required prior to publication. Overall, the paper is a well-written piece of work.
Author Response
- Minor language editing is required prior to publication.
=>It has already been modified.
- Can be improved:Is the article adequately referenced?
=>It has been improved. Some references have been added.
- Can be improved:Are the conclusions thoroughly supported by the results presented in the article or referenced in secondary literature?
=>It has been improved. The conclusion is the author's analysis and thinking based on the validation results.Added some literature to support it.

Reviewer 3 Report
In general, an interesting study. It does suffer from a poor conceptualizing of social capital, which needs to be improved. Also, some language related problems.
“If employees face continuous abusive supervision, their silence will lead to ego depletion”. What does that mean?
“In the short term, employees may use pressure to save their dignity, improve their 43 performance, and reduce the degree of abuse inflicted by their leaders.” How do they save their dignity? I don’t understand.
“The concept of social capital was first proposed by Bourdieu [33]. He insisted that 132 social capital was “the sum of social connections owned by individuals or organizations” 133 and was originally used to describe a resource embedded in personal relationships”
First, Bourdieu was not the first to suggest the concept. Secondly, it should be made clear that with Bourdieu, social capital refers to power-relations. The way it is presented here, it seems to indicate just social network, not the capital of it. With Bourdieu everything relates to power, as in: having something someone does not have. Thus, social capital means with him: knowing people someone else doesn’t know. Because if you know someone high-up in a company and someone does not, you have more social capital than that person. So, it is not just a sum – that is a very basic approach to social capital that Coleman and such took, thereby missing the important power-relations that are inherent to social relations.
“The current research on the impact of social capital is still in its infancy.”
That depends what type you are referring to. There is quite a lot of Bourdieu-research.
“Studies have shown that organization culture has an impact on social capital [35]. In the 137 tourist business, the more hotel employees are aware of social capital, the less eager they 138 are to leave [36]. Some studies have explored social capital as a mediating variable, while 139 few have considered it as a moderating variable..”
It is not clear what definition the authors are then using. What is social capital for the authors in this study? Furthermore, the sentence is somewhat odd. How are hotel employees aware of social capital? Why are they then less eager to leave? (why should they leave anyway?)
The method section is really too short. I have no idea how the study was conducted. More information is needed on the scales. Were they all 5-point scales? What range did each scale have? What were the alpha Cronbach scores?
Line 285 to 286 are quite odd.
The way social capital is operationalized more or less as a form of social support. It is not clear if it really is a social capital measure.
Figure 2 needs marks for axis.
Author Response
- “If employees face continuous abusive supervision, their silence will lead to ego depletion”. What does that mean?
=>A more accurate description was used.
- “In the short term, employees may use pressure to save their dignity, improve their 43 performance, and reduce the degree of abuse inflicted by their leaders.” How do they save their dignity? I don’t understand.
=>A more accurate description was used.
- “The concept of social capital was first proposed by Bourdieu [33]. He insisted that social capital was “the sum of social connections owned by individuals or organizations” and was originally used to describe a resource embedded in personal relationships”
First, Bourdieu was not the first to suggest the concept. Secondly, it should be made clear that with Bourdieu, social capital refers to power-relations. The way it is presented here, it seems to indicate just social network, not the capital of it. With Bourdieu everything relates to power, as in: having something someone does not have. Thus, social capital means with him: knowing people someone else doesn’t know. Because if you know someone high-up in a company and someone does not, you have more social capital than that person. So, it is not just a sum – that is a very basic approach to social capital that Coleman and such took, thereby missing the important power-relations that are inherent to social relations.
=>The literature review section of social capital was revised, consulted more classic literature. The relevant definition is made clear.
- “The current research on the impact of social capital is still in its infancy.”
That depends what type you are referring to. There is quite a lot of Bourdieu-research.
=>A more accurate description was used.
- “Studies have shown that organization culture has an impact on social capital [35]. In the tourist business, the more hotel employees are aware of social capital, the less eager they are to leave [36]. Some studies have explored social capital as a mediating variable, while few have considered it as a moderating variable.”
It is not clear what definition the authors are then using. What is social capital for the authors in this study? Furthermore, the sentence is somewhat odd. How are hotel employees aware of social capital? Why are they then less eager to leave? (why should they leave anyway?)
=>The relevant definition is made clear. We used a more accurate description of this section.
- The method section is really too short. I have no idea how the study was conducted. More information is needed on the scales. Were they all 5-point scales? What range did each scale have? What were the alpha Cronbach scores?
=>The description of the Likert scale has been added.
Cronbach α Coefficients are greater than 0.8, in the diagonal of the table 2.
More content was added to show how the research was conducted.
- Line 285 to 286 are quite odd.
=>It has been adjusted.
- The way social capital is operationalized more or less as a form of social support. It is not clear if it really is a social capital measure.
=>The relevant definition is made clear.
- Figure 2 needs marks for axis. =>It has been modified.
- Must be improved:Is the content succinctly described and contextualized with respect to previous and present theoretical background and empirical research (if applicable) on the topic?
=> It has been improved. The relevant definition is made clear. The theoretical background is reinforced.
- Must be improved:Are the research design, questions, hypotheses and methods clearly stated? =>It has been improved. More detailed instructions have been added.
- Can be improved:For empirical research, are the results clearly presented? =>Improvements have been made.
